# Central catheter tip migration in critically ill patients

**Roei Merin**[1]*, **Amir Gal-Oz**[1], **Nimrod Adi**[1], **Jacob Vine**[1], **Reut Schvartz**[1], **Reut Aconina**[2], **Dekel Stavi**[1]

1 Department of Anesthesiology and Intensive Care, Tel Aviv Sourasky Medical Center, Tel Aviv, Israel,
2 Dept of Medical Imaging, Sunnybrook Health Sciences Centre, Toronto, Canada

* R.merin@gmail.com

## Abstract

### Objectives

Chest X-ray (CXR) is routinely required for assessing Central Venous Catheter (CVC) tip position after insertion, but there is limited data as to the movement of the tip location during hospitalization. We aimed to assess the migration of Central Venous Catheter (CVC) position, as a significant movement of catheter tip location may challenge some of the daily practice after insertion.

### Design and settings

Retrospective, single-center study, conducted in the Intensive Care and Cardiovascular Intensive Care Units in Tel Aviv Sourasky Medical Center 'Ichilov', Israel, between January and June 2019.

### Patients

We identified 101 patients with a CVC in the Right Internal Jugular (RIJ) with at least two CXRs during hospitalization.

### Measurements and results

For each patient, we measured the CVC tip position below the carina level in the first and all consecutive CXRs. The average initial tip position was 1.52 (±1.9) cm (mean±SD) below the carina. The maximal migration distance from the initial insertion position was 1.9 (±1) cm (mean±SD). During follow-up of 2 to 5 days, 92% of all subject's CVCs remained within the range of the Superior Vena Cava to the top of the right atrium, regardless of the initial positioning.

### Conclusions

CVC tip position can migrate significantly during a patient's early hospitalization period regardless of primary location, although for most patients it will remain within a wide range

**Data Availability Statement:** All relevant data are within the paper and its Supporting information files.

**Funding:** The author(s) received no specific funding for this work.

**Competing interests:** The authors have declared
that no competing interests exist.

of the top of the right atrium and the middle of the Superior Vena Cava (SVC), if accepted as
well-positioned.

## Background

Central Venous Catheterization (CVC) is a common procedure in intensive care units (ICU),
operating rooms (OR), and other hospital departments such as the emergency department or
in other cases where peripheral venous access is difficult to obtain (e.g. oncology or hematol-
ogy). It is estimated that in the United States alone 5 million CVCs are inserted each year [1,
2]. In most cases within the ICU, a routine chest X-Ray (CXR) is obtained after insertion in
order to detect insertion-related complications and to assess catheter tip position. In most
cases using the CVC is allowed only after the CXR is conducted.

 With the increasing use of ultrasound (US) in guiding CVC insertion as opposed to ana-
tomical landmarks alone, there has been a decrease in insertion-related complications, mainly
pneumothorax, and vascular bleeding [3, 4]. In addition, the use of bedside-US has been
shown to be faster and more accurate compared to CXR in detecting these complications
when they occur [2, 5, 6]. For these reasons, currently, the assessment of CVC tip position
remains the main rationale for routine CXR before usage when performing US assisted CVC
insertion.

 Malposition of the tip can potentially lead to CVC-related complications. Proximal posi-
tioning has been related to a higher risk of venous thrombosis, while distal positioning in the
right atrium or in the right ventricle can cause cardiac arrhythmias and tamponade, although
late cardiac tamponade as a complication of deep catheter tip location has been described as
an "urban legend" [1, 7–9]. Nevertheless, proper positioning of CVC tip is a subject of contro-
versy in the literature and is based on outdated guidelines [10]. Several past studies accept only
a narrow area for proper tip positioning, usually in the lower part of the Superior Vena Cava
(SVC)- an area of 2–3 cm long below the Carina (see zone A in Fig 1) [1, 6, 11, 12], while oth-
ers allow a much larger area for tip positioning: from the upper part of the SVC to the right
atrium [10, 13–19] (Zone A and B as depicted in Fig 1). Whether a tip located in the right
atrium (RA) is accepted is also a matter of controversy [10, 11]. Regardless of tip position after
insertion, the CVC is often left unchanged due to the likelihood of complications associated
with CVC repositioning [19, 22].

 Although there is much discussion regarding assessment of post insertion primary tip loca-
tion, the migration of tip position during hospitalization is not routinely examined and has
not been well described in previous literature. The importance of tip position migration in the
following days post insertion is unknown and it is not clear how this issue should be addressed,
if at all.

 This study aims to determine the migration of CVC tip location during hospitalization, as
significant movement can challenge traditional approach.

## Material and methods

This is a retrospective observational study conducted in the Intensive Care and Cardiovascular
Intensive Care Units in Tel Aviv Sourasky Medical Center 'Ichilov', Israel. We examined all
patients from our EMR (electronic medical records) hospitalized between January to June
2019 in the general Intensive Care Unit (ICU) and Cardiovascular Intensive Care Unit
(CVICU). In both units, a 15 cm long CVC is routinely inserted using a US-guided technic
by anesthesiologists and intensive care physicians all trained in the same critical care and anes-
thesia program. Patients were included if they were aged over 18, had their CVC inserted from

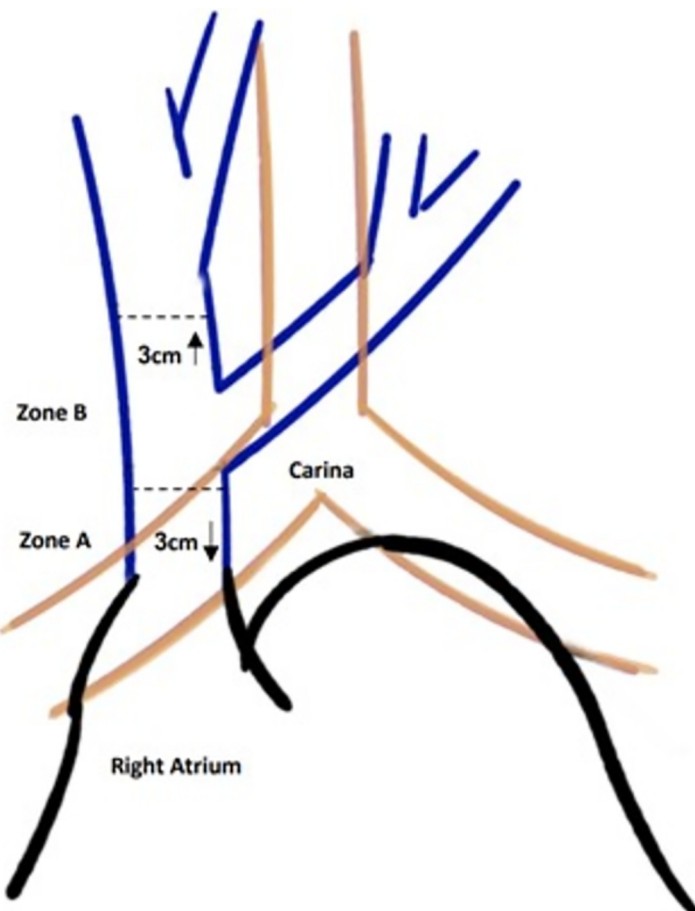

**Fig 1. Relationship between the Superior Vena Cava and the carina.**

the Right Internal Jugular (RIJ), and had at least two CXRs showing the CVC's position during their hospitalization. Only RIJ catheters were included in order to minimize diversity regarding insertion sites. We identified 162 patients who met the inclusion criteria. Patients under the age of 18, hospitalized for less than 24 hours, patients in a prone position, pregnant individuals, and patients undergoing dialysis were excluded from the study. In addition, we excluded patients who underwent another cardio-vascular or thoracic surgery or moved between wards during the time of the study in order to minimize these effects on CVC location. The ethical committee has waived the need for informed consent.

For each patient, the time of CVC insertion was noted. We then analyzed the post-procedure CXR to determine the primary catheter tip position. Using "Vue Motion" software (PHILIPS, Version 12, software for picture archiving and communication) the vertical distance between the carina and the tip location was used as a standardized method to describe the position of the catheter tip on CXR (as shown in Fig 2). Next, we applied the same method on every following CXR performed while the CVC was still inserted and untouched, up to 4 CXRs for each patient. In our facility in the early days of intensive care hospitalization, a CXR is performed every one or two days, so a follow-up time of up to 5 days from CVC insertion was selected. CXRs were examined by two different ICU physicians/trainees who were qualified for this routine procedure: R.M and J.V. In any case of disagreement up to 1 cm, the

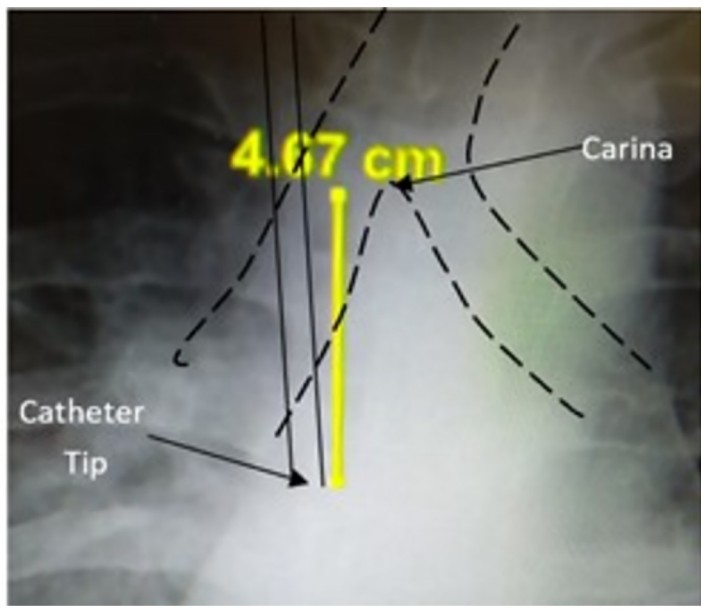

**Fig 2. Measuring the vertical distance from carina to catheter tip.**

average between the two investigators was used. In any other case, a third investigator (D.S) examined the CXR.

In all cases, we noted CVC insertion time, time of each CXR, patient's position during CXR (supine/sitting/standing), department in which CVC was inserted (ICU/CVICU), and patient's demographic data (sex, age, height, weight).

We used the definition for 'correct' CVC tip position as the lower part of the SVC, 1–3 cm below the carina, based on guidelines published in 2016 [20]. Catheter tip positioned inside this range on CXR was defined as well-positioned, otherwise, it was defined as malpositioned. We calculated the average movement between two consecutive CXRs and the maximal distance between the initial tip position and the tip position on follow-up CXRs.

All data were analyzed using SPSS version 21.0. Continuous data are described as mean ± SD (standard deviation) and categorical variables are given as no. (%) continuous variables were compared using a two-sided T-test, P_value <0.05 was considered to be statistically significant. Data was proved for normal distribution before performing T-test. Multivariant analysis was performed using linear regression as the main outcome variable is catheter tip migration. Change in CVC position (well-position or malposition) between follow up CXRs was analyzed in a descriptive approach using percentage of CVCs compared to initial position.

## Results

We identified 162 patients who had a CVC inserted in the study period. 13 patients (9%) had only one CXR showing the CVC position. Out of the remaining 149 patients, 48 (29%) had a CVC inserted in a position different than the RIJ. 101 patients met the inclusion criteria for the study. All CVCs were inserted according to the institutional guidelines and were secured similarly using skin stitches and dressing. Demographic data are shown in Table 1.

In 100 patients of a total of 101 patients (99%) the CVC was inserted into the SVC, and 1 CVC (1%) was inserted from the RIJ and ended in the Right Subclavian. For the remaining 100 patients, CVC position was measured in the initial CXR and was followed in the

**Table 1. Patient characteristics, compared between initially well-positioned and malpositioned CVCs according to CXR after insertion.**

| | Total (n = 101) | well-positioned Catheter tip in the first CXR (n = 46) | malpositioned Catheter tip in the first CXR (n = 55) | P.Value |
|---|---|---|---|---|
| **Age, years mean (SD)** | 63.1 (11.3) | 64.04(10.1) | 62.38(12.2) | 0.45 |
| **Female sex, n (%)** | 25 (25%) | 10 (22%) | 15(27%) | 0.48 |
| **Height, m mean (SD)** | 1.7 | 1.71 (0.09) | 1.69 (0.07) | 0.086 |
| **Weight, Kg mean (SD)** | 79.96 (18.97) | 79.1 (16.32) | 80.1 (20.9) | 0.28 |
| **BMI mean (SD)** | 27.7(5.9) | 27.8(6.6) | 27.6(5.1) | 0.18 |
| **Unit** | | | | |
| **ICU, n (%)** | 51 | 21 (41%) | 30 (59%) | 0.315 |
| **CVICU, n (%)** | 50 | 25 (50%) | 25(50%) | 0.5 |

CXR- Chest X-ray; SD- standard deviation; BMI- Body Mass Index; ICU- Intensive Care Unit; CVICU- Cardiovascular Intensive Care Unit.

consecutive CXR during hospitalization: 12 patients had one follow up CXR before CVC extraction, 41 patients had 2 CXR's and 47 had 3 CXRs follow up. The average follow up time was 2.6 days (range 2–5 days). Average time from insertion to first CXR was 3 hours. In the first CXR assessment after CVC insertion, 46 CVC tips were positioned inside the lower part of the SVC (well-positioned), and 54 CVC tips were positioned either deeper or higher than that range (malpositioned). Of the 46 CVCs that were initially well-positioned, 22 (48%) remained well-positioned in the second CXR and only 8 (17%) remained well-positioned in all CXRs during hospitalization. In 15% of CXR's defining the tip position was difficult due to reduced x-ray quality, and additional examination by ICU specialist was needed. Table 2 shows the follow-up of initially well positioned and initially malposition CVCs according to the 2nd 3rd and 4th CXRs.

## Catheter location movement during hospitalization

The initial average CVC position was 1.5 (±1.9) cm (mean±SD) below the Carina level. The highest position was 4.6 above, and the deepest was 5.5 cm below the Carina level. The average movement of CVC between two consecutive CXRs (around 24h) was 1.1 cm (±0.7) (mean±SD), and the maximal distance from the initial insertion position was 1.9 (±1) cm (mean±SD). There were no significant differences in CVC movement comparing initially well-positioned and malpositioned catheters (1.12 for wellpositioned and 1.08 for malpositioned, p_value 0.4).

**Table 2. Catheter tip positioning movement during hospitalization as shown in consecutive CXRs compared to initial position (1st CXR).**

| | Total | Well-position of catheter tip | Malposition of catheter tip |
|---|---|---|---|
| **1st CXR Well positioned Catheter tip** | 46 | 100% | |
| **2nd CXR** | 46 | 22(48%) | 24 (52%) |
| **3rd CXR** | 41 | 20 (49%) | 21 (51%) |
| **4th CXR** | 27 | 11(40%) | 16(60%) |
| **1st CXR malpositioned Catheter tip** | 54 | | 100% |
| **2nd CXR** | 54 | 21(39%) | 33(61%) |
| **3rd CXR** | 47 | 12(26%) | 35(74%) |
| **4th CXR** | 20 | 10(50%) | 10(50%) |

CXR- Chest X-ray.

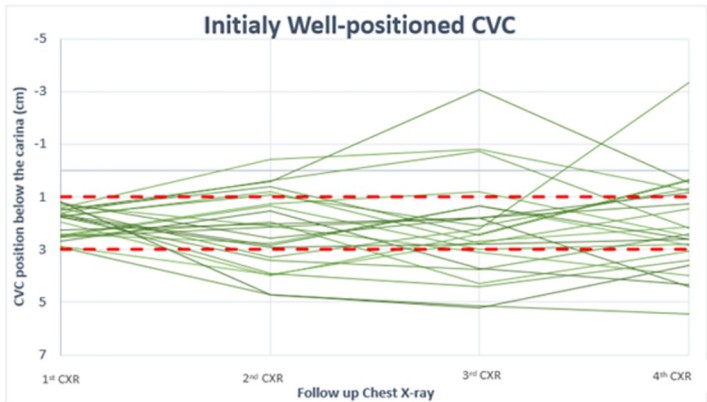

**Fig 3. Tip position below the carina as shown in CXR (after insertion, 1st, 2nd, and 3rd) for initially well-positioned catheters.** CVC (Central Venous Catheter), h(Hour). CVC- Central Venous Catheter.

Figs 3 and 4 present the movement of CVC tip during hospitalization for patients with four CXRs, a total of 47 patients. The "well positioned" zone, between 1–3 cm below the carina, is shown in red. Fig 3 presents patients with CVC who were initially well positioned, and Fig 4 presents patients initially malpositioned. At each follow-up CXR, the chance of an initially well-positioned and initially malpositioned catheter to be in the "well positioned" zone was as follows: 48% vs 39% at first CXR, 49% vs 26% at second, and 40% vs 50% at third CXR.

When assessing tip location using a wider range for well-positioning, Figs 3 and 4 above show that most catheters regardless of initial position, remain between 1.5 cm above the carina to 5 cm under it. Only four CVCs (8%) were documented out of this range during use. All of these were in patients with BMI> 40 or height<1.55 m.

In a multi-parameters regression model (Table 3), we assessed for parameters related to CVC movement. Patient's BMI (P_value = 0.03) and patient's position change between CXRs (P_vaule = 0.04) were both statistically significant in correlation to CVC tip movement. Patient's age (P_value = 0.08) and sex (P_value = 0.4) did not affect the average movement.

Table 3 shows the average movement of CVC tip from initial location depending on the patient's BMI and change in position during hospitalization using the linear regression model.

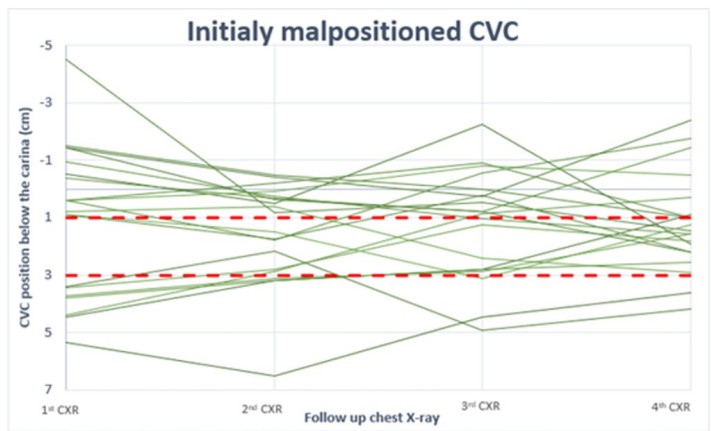

**Fig 4. Tip position below the carina as shown in CXR (after insertion, 1st, 2nd, and 3rd) for initially mal-positioned catheters.** CVC (Central Venous Catheter), h(Hour).

**Table 3. Multiparameter regression analysis of CVC tip movement between two consecutive CXRs.**

| BMI | Position change | Mean movement | (min) = | (max) = |
|---|---|---|---|---|
| 20 | no | 1.083 | .198 | 1.968 |
| 20 | yes | 1.377 | .205 | 2.548 |
| 25 | no | 1.187 | .206 | 2.167 |
| 25 | yes | 1.481 | .214 | 2.747 |
| 30 | no | 1.291 | .215 | 2.367 |
| 30 | yes | 1.584 | .223 | 2.946 |

When comparing tip position and movement for CVCs inserted in the ICU and the CVICU departments, a significant difference in the initial tip position was identified with an average of- 2.1 cm below the carina in ICU patients vs 0.9 cm below the carina in CVICU patients (P_value = 0.001). In the ICU, 41% of CVCs were initially well-position compared to 50% in the CVICU (P_value 0.315), while both units use the same 15 cm long CVC. No significant change in tip migration was identified (1.95 cm for ICU and 1.85 cm for CVICU, P_value 0.64).

No CVC-related complications were recognized in the post insertion CXR and none of the CVCs were repositioned after the first insertion.

## Discussion

Although CVC insertion is a common procedure, an optimal CVC tip location definition is still a matter of debate that vary between recommendations [10]. Most literature and recommendations discuss initial positioning of the CVC, without addressing the indications and risks for repositioning. Moreover, CVC tip location immediately post insertion and during different stages of its use are of the same meaning, but knowledge of the tip migration, it's significance and the need for follow up, lacks.

In this study, we set to examine the movement of CVC tip throughout its use. We examined both the range of movement between each two consecutive CXRs (usually a day apart), and the likelihood of CVC tip to remain within a narrow range as recommended in previous literature [1, 11]. We have learned that 1) after insertion, about half the catheters were outside the "narrow" range. 2) There was a significant average movement of 1.9 (±1) cm (mean±SD) of the CVC tip from initial position and 3) Initial optimal CVC location cannot predict correct positioning of CVC during the first days of hospitalization.

Based on these results, targeting a narrow range as a desirable catheter position [1, 11] may be challenging, as catheters will migrate in and out of that zone, while the significance and the need to manage that are questionable. Should a wider range approach be taken [13–15], tip location will most likely be initially located and remain within that zone. Regarding the safety of using a large range for CVC positioning, a relationship has been shown between CVC position and catheter-related thrombosis, tamponade, and arrhythmias, but these are rare complications, mainly related to catheters positioned higher than the SVC or deep within the right atrium [1, 18]. Following insertion, assessment of the tip position using CXR is widely used in Intensive Care Units as a mandatory practice (although different recommendations for operating theatre settings exist [21]). This method, although commonly used, has been repeatedly questioned [7, 19]. With the growing use of US, CVC positioning remains the main indication for performing CXR prior to its use [7, 22]. For all of these issues, our data shows that using a 15cm long catheters with US-guided RIJ approach will result in most cases (adults with BMI<40 and height >155 cm) in positioning of the CVC within the SVC or at the upper part of the right atrium and will remain there during hospitalization.

Routine CXR for CVC assessment as a mandatory step before initiating it's use, potentially affects patient care, mainly by delaying care [6, 19]. Extreme ICU environment such as the COVID pandemic can potentially increase delays. In addition, there are contradicting recommendations as to the necessity of action when a nonoptimal CVC tip is diagnosed, as repositioning of the CVC can lead to unnecessary patient discomfort and more complications (mainly infection), which may be more common and severe than complications from suboptimal tip position [23, 24]. Because of the relatively low incidence of complications, our study was not large enough nor was it targeted to discuss safety ((in our study no adverse effects were recorded), and further studies with larger cohorts are needed. Nevertheless, a wider range is well described in the literature [8].

In this study, we did not find a precise way to predict the movement of CVCs during their use. Although BMI was found to be a significant factor in CVC tip movement, it provides only a partial explanation of the diversity between patients, as other factors such as the X-ray angle and patient's position might be involved. It has also been shown that a 1–5 cm movement of tip location can be related to patient's head maneuvers [25]. Alternative methods of demonstrating tip position using US have been described [6, 26], though these methods are not routinely used.

Given the high probability that CVC tip will be within the SVC or at the upper part of the right atrium together with the unpredictability of CV movement during hospitalization, assessing the exact initial positioning after insertion has limited significance. Therefore, in selective cases when use of CVC is urgent, routine CXR might be used as earliest as possible but without delaying treatment.

In our facility, CVC insertion is performed mostly in two different settings; in the ICU, CVCs are inserted bedside, using a US-guided technique, by a trained physician and a routine CXR is mandatory before the use of the CVC. In the operation theatre, CVCs are inserted (before heart surgeries etc.), by similarly trained physicians, but CXR is performed only after surgery is done, in the CVICU (hours after insertion). When comparing between ICU and CVICU, there was a significant difference in CVC position post insertion, while movement of the tip during hospitalization was similar. Differences in the position between these scenarios may be secondary to the different time interval and manipulations between insertion and CXR. Because the difference in the initial tip position is less than the average tip migration, these differences are probably without any clinical importance.

Our study's limitations relate to the nature of CXR performed in the ICU/CVICU units; using a mobile x-ray machine while the patient is in a supine position can lead to reduced CXR quality and difficult analysis. This was the case in around 15% of CXRs observed and an additional examination by an ICU attending was needed. Another limitation of this study is the variability in tip positioning caused by the patient's head position during CXR and anatomical variants which were not measured during the study. These factors and limitations exist in the "every day" ICU settings and present a challenge for the clinician in every tip position assessment after CVC insertion, therefore they do not weaken this study's conclusions, but rather reflect the daily routine. Another limitation was that CVCs were inserted by different physicians from different departments. We believe this had only a little effect, as all were ICU or Anaesthesia physicians with similar training working under the same guidelines. Finally, the size of the study group did not allow us to assess the rate of complications and relationship to different locations of the catheter tip or its migration, a larger study is needed to address this issue.

## Conclusions

CVC tip position migrates throughout its use, thus an initial optimal position within a narrow range does not reflect or predict its position later during treatment. Should a wider range of tip

position be accepted, a 15 cm CVC inserted through the RIJ in most adults with height over 155 cm and BMI<40 will initially reside within the SVC or top right atrium and remain in that location throughout its use.

## Supporting information

**S1 Data. Publication data.**
(XLSX)

## Author Contributions

**Conceptualization:** Nimrod Adi, Dekel Stavi.

**Data curation:** Roei Merin, Jacob Vine.

**Formal analysis:** Reut Schvartz.

**Investigation:** Amir Gal-Oz, Reut Schvartz, Reut Aconina.

**Methodology:** Roei Merin, Amir Gal-Oz, Nimrod Adi, Dekel Stavi.

**Project administration:** Nimrod Adi.

**Supervision:** Nimrod Adi, Dekel Stavi.

**Writing – original draft:** Roei Merin, Dekel Stavi.

**Writing – review & editing:** Roei Merin, Amir Gal-Oz, Jacob Vine, Reut Aconina, Dekel Stavi.

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
