## [Decision Letter · Decision Letter 0]

14 Jul 2022

PONE-D-22-14765Assessing Central Venous Catheter tip location- a different approach?PLOS ONE

Dear Dr. Merin,

Thank you for submitting your manuscript to PLOS ONE. After careful consideration, we feel that it has merit but does not fully meet PLOS ONE’s publication criteria as it currently stands. Therefore, we invite you to submit a revised version of the manuscript that addresses the points raised during the review process.

We look forward to receiving your revised manuscript.

Kind regards,

Martin Kieninger

Academic Editor

PLOS ONE

Journal Requirements:

Reviewers' comments:

Reviewer's Responses to Questions

**Comments to the Author**

1. Is the manuscript technically sound, and do the data support the conclusions?

Reviewer #1: No

Reviewer #2: Yes

2. Has the statistical analysis been performed appropriately and rigorously? 

Reviewer #1: No

Reviewer #2: Yes

3. Have the authors made all data underlying the findings in their manuscript fully available?

Reviewer #1: No

Reviewer #2: Yes

4. Is the manuscript presented in an intelligible fashion and written in standard English?

Reviewer #1: Yes

Reviewer #2: Yes

5. Review Comments to the Author

Reviewer #1: Dear authors,

thank you for submitting your article and data. I believe that the article needs some major revision to clarify the message for the readers. I hope that my suggestions help you to improve your article.

Major concerns:

The title of your work suggests that you present a new/different approach to controlling catheter location with x-rays. But, you do not! Instead, you investigate catheter migration over time with consecutive x-rays. Choose an appropriate title.

In the discussion section, you write: …..examine the movement of CVC tip throughout its use. Exactly, this should be reflected in the title.

Follow up is 72 hours? Why? You state that every x-ray was evaluated in the method section. Why did you only use a 72 hour follow up as mentioned in the abstract and figure 3 and 4?

Material and Method

Clarify, how patients were included into the study. Do you have a database. Did you screen your PDMS, etc. for every patient admitted between xxx and xxx?

The exclusion criteria are documented sufficiently.

How was the mentioned distance between carina and catheter tip measured? Did you use a software? Did you use a x-ray fluoroscope/screen? A difference between tow investigators of more than 1 cm is mentioned – this seems large using a software?

Who are the investigators? Part of your study team or part of the wards? What was there training? How did the training of the ICU specialist differ how was consulted in case of poor x-ray quality? Did you check with radiology?

Statistics. You used a T-test. Did you check for normal distribution? If data does not show normal distribution a non-parametric test should be used.

The multi-parameters regression model used in table 3 has to be explained in the method section.

Results and Discussion

Major influencing factors of possible catheter mobilizations are not investigated or mentioned; e.g. prone positioning, mobilization of patients, insertion duration of the catheters, etc. If these data are not available, it should at least be discussed as limitation.

Adverse events/e.g. the non-existence of adverse events is not mentioned or discussed. Not one catheter had to be removed (even the one in the subclavian vein)?

Conclusions

You only included 101 patients. Relativize your conclusion: ….. a 15 cm CVC inserted through the RIJ in most adults will likely initially reside ….. Refer to the height and weight of the studied patients.

General

I suggest to revise the paper. Focus on your major statement: catheters may move by several centimeters of the course of time (with or without major adverse events???).

Include data to these observed/non-observed adverse events? After all, what does the catheter movement mean for our patients/clinical practice?

Minor concerns

Abstract:

Design and Settings: Retrospective, single-center study, conducted in the Intensive Care and Cardiovascular Intensive Care Units between January and June 2019….

- State the name of the hospital/institution here including city and country

The average initial tip position was 1.52 (±1.9)…..

- Clarify, you probably refer to mean ± SD

Central Venous Catheterization (CVC) is a common procedure in intensive care units (ICU),

operating rooms (OR), and other hospital departments.

- Specify ‘other hospital departments’ or give examples ‘other departments like…..’ or delete.

….a routine chest XRay (CXR) is obtained after insertion and before using the CVC in order to detect insertion related complications and to assess catheter tip position (CTP).

- Really? One x-ray after insertion and a new one before use? How long do you wait until you use newly inserted CVCs in your ICU?

In addition, the use of bedside-US has been shown to be faster and more accurate in detecting these complications when they occur.

- Specify, faster than what else (x-rays)?

CTP

- You already use CVC as abbreviation. I suggest to use only CVC and refer to the ‘tip of the CVC’, since both abbreviations are similar.

….performing US assisted or guided CVC insertion

- Clarify. What is the difference?

…m the upper part of the SVC to the right atrium10,13–19 (Zone A and B).

- Clarify, Zone A and B as depicted in figure xxx.

All data were analysed by SPSS version 21.0.

- Suggestion: All data were analysed using SPSS version 21.0.

At each follow-up CXR, the chance of an initially well-positioned and initially malpositioned catheter to be in the "safezone" were similar (48% vs 39% at first CXR, 49% vs 26% at second and 40% vs 50% at third CXR.

- Do not use ‘safezone’, use the terminology introduced above. You refer to similar, but I do not see a statistical evaluation. If a descriptive approach was used it has to be explained in the method section.

We examined both the range of daily movement…

- No! Do you perform routine daily x-rays on your ICU? If so, explain in the method section. This would contribute to the work. Explain, when follow up x-rays were taken.

There was a significant average movement of was 1.9 (±1) cm of CTP and thus predicting accurately the tip position (within a narrow range) is limited.

- Does not make sense. Clarify. Of course, one cannot predict a future positioning because of a current exact position.

Reviewer #2: This manuscript is a retrospective study on assessing the migration of the tip of IVC in hospitalized patients. The authors have concluded that the tip of CVC can migrate throughout patients’ hospitalization course, but “will remain within a wide range of the top of the right atrium and the middle of the Superior Vena Cava (SVC), if accepted as well-positioned”.

In terms of the originality of the article, there have been descriptions of several cases of catheter migration, but I could not find any prospective or retrospective studies in this area.

The title of this manuscript could be changed, it should be more explanatory of the study itself. Please, consider changing the title. The abstract portion of the manuscript clearly reflects its content.

In results in Table 2, please specify the descriptions. If catheter was malpositioned since initial placement, how you can make a column of “Well-positioned catheter”. Consider revising.

In the discussion, you should also add several limitations of the study secondary to its retrospective origin:

1. Even though catheters were placed by two teams of providers, there is a necessity to mention that difference in providers skills should be a consideration

2. There was no mention of how the CVCs were secured in place. That also might add to a higher or lower possibility of migration, depending on how CVCs were secured and dressings applied.

In conclusion, I would recommend this Manuscript for publication with the change of the title and minor correction as I mentioned above.

6. PLOS authors have the option to publish the peer review history of their article (what does this mean?). If published, this will include your full peer review and any attached files.

Reviewer #1: No

Reviewer #2: **Yes: **Eugenia Ayrian, MD

---

## [Author Response · Author response to Decision Letter 0]

10 Aug 2022

This is a revised version of our manuscript after addressing and correcting the reviewers’ extensive and constructive comments.

Attaches is a point-by-point response and description of all changes to the manuscript (a version of the manuscript with highlighted changes is also attached).

---

## [Decision Letter · Decision Letter 1]

8 Sep 2022

PONE-D-22-14765R1Central catheter tip migration in critically ill patientsPLOS ONE

Dear Dr. Merin,

Thank you for submitting your manuscript to PLOS ONE. After careful consideration, we feel that it has merit but does not fully meet PLOS ONE’s publication criteria as it currently stands. Therefore, we invite you to submit a revised version of the manuscript that addresses the points raised during the review process.

We look forward to receiving your revised manuscript.

Kind regards,

Martin Kieninger

Academic Editor

PLOS ONE

Journal Requirements:

Reviewers' comments:

Reviewer's Responses to Questions

**Comments to the Author**

1. If the authors have adequately addressed your comments raised in a previous round of review and you feel that this manuscript is now acceptable for publication, you may indicate that here to bypass the “Comments to the Author” section, enter your conflict of interest statement in the “Confidential to Editor” section, and submit your "Accept" recommendation.

Reviewer #1: All comments have been addressed

Reviewer #2: (No Response)

2. Is the manuscript technically sound, and do the data support the conclusions?

Reviewer #1: Yes

Reviewer #2: Yes

3. Has the statistical analysis been performed appropriately and rigorously? 

Reviewer #1: Yes

Reviewer #2: Yes

4. Have the authors made all data underlying the findings in their manuscript fully available?

Reviewer #1: Yes

Reviewer #2: Yes

5. Is the manuscript presented in an intelligible fashion and written in standard English?

Reviewer #1: Yes

Reviewer #2: Yes

6. Review Comments to the Author

Reviewer #1: Dear authors,

thank you for accepting the suggestions.

Some of the revised scentences are hard to read:

Another limitation rests with the retrospective origin of the trail- CVC's were inserted

by different physicians, and although all were ICU or Anaesthesia physicians with similar

training working under the same guidelines, of nature each have different skill that were not

taken into account.

-> Please, consider to break it down.

Some typos 'sneaked' into the revised version:

... with hight over 155 cm and BMI<40....

-> Should be: height

The article needs to be checked for typos and akward phrasing before publication.

Reviewer #2: Please make minor changes to the manuscript:

Materials and methods: -"Patients under the age of 18, hospitalized less than 24 hours, prone position, pregnant individuals..." - Add ‘Patients in' prone position...

-“After proving normality“ - please revise

Table 2: when describe the “Malpositioned CVCs” you have to change the name of “Well-positioned tip”, since there is no well-positioned tip in mal-positioned catheters. You possibly mean that the catheter tip was not migrated from initial malposition, but it is not well understood from the description - revise.

7. PLOS authors have the option to publish the peer review history of their article (what does this mean?). If published, this will include your full peer review and any attached files.

Reviewer #1: No

Reviewer #2: No

---

## [Author Response · Author response to Decision Letter 1]

5 Oct 2022

We would like to thank you again for the constructive comments. All of the reviewer's comments were asnwered and corrected as shown in the attached respons letter. 

We believe the manuscriped had improved due to the reviewers’ thoughtful and constructive comments and suggestions for revision.

---

## [Decision Letter · Decision Letter 2]

1 Nov 2022

Central catheter tip migration in critically ill patients

PONE-D-22-14765R2

Dear Dr. Merin,

We’re pleased to inform you that your manuscript has been judged scientifically suitable for publication and will be formally accepted for publication once it meets all outstanding technical requirements.

Kind regards,

Martin Kieninger

Academic Editor

PLOS ONE

Additional Editor Comments (optional):

Reviewers' comments:

Reviewer's Responses to Questions

**Comments to the Author**

1. If the authors have adequately addressed your comments raised in a previous round of review and you feel that this manuscript is now acceptable for publication, you may indicate that here to bypass the “Comments to the Author” section, enter your conflict of interest statement in the “Confidential to Editor” section, and submit your "Accept" recommendation.

Reviewer #1: All comments have been addressed

Reviewer #2: All comments have been addressed

2. Is the manuscript technically sound, and do the data support the conclusions?

Reviewer #1: Yes

Reviewer #2: Yes

3. Has the statistical analysis been performed appropriately and rigorously? 

Reviewer #1: Yes

Reviewer #2: Yes

4. Have the authors made all data underlying the findings in their manuscript fully available?

Reviewer #1: Yes

Reviewer #2: Yes

5. Is the manuscript presented in an intelligible fashion and written in standard English?

Reviewer #1: Yes

Reviewer #2: Yes

6. Review Comments to the Author

Reviewer #1: In the method section you wrote:

Data was proved for normal distribution before performing T-test.

It should be:

Data was tested for normal distribution before performing a T-test. (or: Normal distribution was proven before performing a T-test.)

Reviewer #2: Thank you for accepting my suggestions. I am satisfied with all changes. Manuscript can be published

7. PLOS authors have the option to publish the peer review history of their article (what does this mean?). If published, this will include your full peer review and any attached files.

Reviewer #1: No

Reviewer #2: No

---

## [Editor Report · Acceptance letter]

9 Dec 2022

PONE-D-22-14765R2 

Central catheter tip migration in critically ill patients 

Dear Dr. Merin:

I'm pleased to inform you that your manuscript has been deemed suitable for publication in PLOS ONE. Congratulations! Your manuscript is now with our production department. 

Kind regards, 

on behalf of

Dr. Martin Kieninger 

Academic Editor

PLOS ONE